



# A Multi-phasic Approach for Estimating the Biot Coefficient for Grimsel Granite

Patrick Selvadurai[a,∗], Paul Selvadurai[b,∗∗], Morteza Nejati[b,∗∗]

[a]*Department of Civil Engineering and Applied Mechanics, McGill University, Montréal, QC, Canada H3A 0C3*
[b]*Department of Earth Sciences, ETH Zurich, Switzerland*

**Abstract**

This paper presents an alternative approach for estimating the Biot coefficient for the Grimsel granite, which appeals to the multi-phasic mineralogical composition of the rock. The modelling considers the transversely isotropic nature of the rock that is evident from both the visual appearance of the rock and determined from mechanical testing. Conventionally, estimation of the compressibility of the solid material is performed by fluid saturation of the pore space and pressurization. The drawback of this approach in terms of complicated experimentation and influences of the unsaturated pore space is alleviated by adopting the methods for estimating the solid material compressibility using developments in theories of multiphase materials. The results of the proposed approach are compared with estimates available in the literature.

*Keywords:* Biot coefficient, transversely isotropic rocks, compressibility of the solid materials, Hashin-Rosen estimates, Voigt-Reuss-Hill estimates

## 1. Introduction

The classical theory of poroelasticity proposed by Biot (1941) is a major contribution to the disciplines of geosciences and geomechanics with applications that include porous earth materials saturated by fluids. The studies in this area are numerous and no attempt will be made to provide a comprehensive survey of past and recent developments. Advances in the area of poroelasticity, and its applications to problems in geomechanics in particular are given by Rice and Cleary (1976); Yue and Selvadurai (1995); Selvadurai (1996, 2007); Wang (2000); Verruijt (2015); Cheng (2015); Selvadurai et al. (2015); Selvadurai and Suvorov (2016) and others. The basic development of the classical theory of poroelasticity relies on constitutive assumptions of Hookean elastic behaviour of the porous skeleton and Darcy flow through the porous medium. In addition, an important component of the theory relates to the partitioning of the total stress tensor for the poroelastic solid between the stresses carried purely by the porous skeleton and the stresses carried by the pore fluid. The partitioning is an important component in the theory of poroelasticity that allows the time-dependent shedding of the applied stresses from the pore fluid to the porous skeleton. The stresses sustained by the porous skeleton have important consequences to the definition of failure of the poroelastic material either through the development of damage (Selvadurai, 2004; Selvadurai and Shirazi, 2004, 2005; Selvadurai et al., 2015), or fracture development and boundary effects on heterogeneities (Selvadurai et al., 2011; Selvadurai and Głowacki, 2017, 2018) or plastic flow (Selvadurai and Suvorov, 2012, 2014). From an environmental geosciences perspective, alterations in

---

∗Corresponding author, William Scott Professor and Distinguished James McGill Professor
∗∗Post-Doctoral Fellow
*Email address:* `patrick.selvadurai@mcgill.ca` (Patrick Selvadurai )



the skeletal permeability associated with its damage can lead to enhanced migration of contaminants and hazardous materials. In Biot's theory, the partitioning of the total stress is achieved through consideration of the bulk modulus of the porous skeleton ($K_D$) and the bulk modulus of the solid material composing the porous skeleton ($K_S$), which introduces the Biot coefficient $\alpha$ and for an isotropic elastic skeleton, has the form $\alpha = 1 - (K_D/K_S)$. When the bulk modulus of the solid material is large in comparison to the skeletal bulk modulus, $\alpha \to 0$, which is the conventional stress partitioning approach proposed in the theory of soil consolidation proposed by Terzaghi (1923). Unlike in soils, the Biot coefficient for rocks can be less than unity. If Biot's classical theory of poroelasticity is accepted, values of $\alpha$ cannot be greater than unity. Such a value would imply that either $K_D < 0$ or $K_S < 0$, which would violate the positive definiteness arguments for the strain energy of an elastic porous skeleton (Davis and Selvadurai, 1996; Selvadurai, 2000) with no locked-in self equilibrating stresses (i.e. the skeleton expands under compressive isotropic stresses). A range of values for $\alpha$ is given by Detournay and Cheng (1993); Wang (2000); Cheng (2015).

The experimental procedure for determining the Biot coefficient $\alpha$ involves estimating the bulk modulus of the porous skeleton ($K_D$), which, in the case of an isotropic skeletal fabric, can be obtained by subjecting a dry or moisture free and jacketed specimen of the rock to isotropic compression and measuring the resulting volumetric strain. This is a straightforward experimental technique and the results can also be verified by conducting uniaxial compression tests on the isotropic rock and measuring both the Young's modulus and Poisson's ratio. The measurement of the compressibility of the solid material composing the skeletal fabric can be either straightforward or complicated depending on the permeability characteristics of the porous material. For rocks with relatively high permeability (e.g. Indiana limestone $10^{-13} \sim 10^{-15}\text{m}^2$ (Selvadurai and Glowacki, 2008; Selvadurai and Selvadurai, 2010, 2014), Vosges sandstone $\sim 10^{-13}\text{m}^2$ (Moulu et al., 1997), etc.), the pore space of the rock can be saturated by initiating a combination of steady flow and vacuum saturation. To determine the compressibility of the solid material, the confining isotropic stresses are allowed to nearly equilibrate with the pore fluid pressure and the volume changes measured can be used to estimate the compressibility of the solid material composing the porous fabric. With very low permeability materials (e.g. the Cobourg limestone $\sim 10^{-23}\text{m}^2$ to $10^{-19}\text{m}^2$ (Selvadurai et al., 2011)), the process of saturation of the pore space can take an inordinately long time with no assurance that the entire pore space is fully saturated or that there are no residual pore fluid pressure artifacts (Selvadurai, 2009). Furthermore, even if the pore space is saturated, attaining equalization of the externally applied pressure with the internal pore fluid pressure can take substantial time (for 150 mm diameter cylindrical Cobourg limestone samples, more than 100 days are required for saturation). For this reason, Selvadurai (2018) proposed an alternative approach where the compressibility of the solid material phase(s) can be estimated by considering the multi-phasic theories developed for estimating the effective properties of composite elastic materials. The composite material theories associated with the Voigt-Reuss-Hill estimates (Voigt, 1928; Reuss, 1929; Hill, 1952, 1965) and the upper and lower bound estimates proposed by Hashin and Shtrikman (1963) can be used to estimate the bulk modulus of the solid material (see also Walpole, 1966; Francfort and Murat, 1986). In this paper, we apply these basic concepts to determine the Biot coefficient for the Grimsel granite. This granite is encountered in the Underground Research Laboratory constructed in Grimsel, Switzerland, in order to perform heater experiments to simulate the thermo-hydro-mechanical (THM) loading associated with heat-emitting containers in the event that the site is chosen as a repository for the deep geologic disposal of high level nuclear fuel waste (i.e. the Full-scale Engineered Barriers EXperiment (FEBEX).) A typical section along the Grimsel Laboratory associated with the FEBEX heater experiment location is shown in Figure 1.

The Aar granitic rock (also referred to as Aare granitic rock) setting at Grimsel has been associated with initiatives related to the use of granitic rock formations as potential hosts for the creation of deep geologic repositories for the



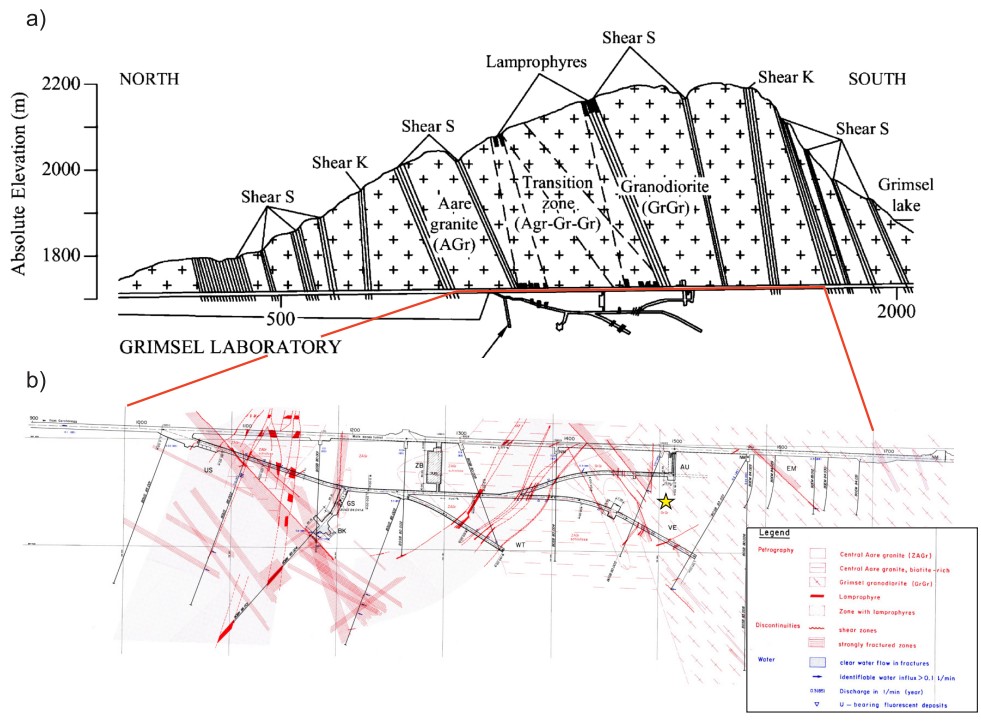

Figure 1: (a) The Grimsel Laboratory and the FEBEX Drift [After Alonso et al. (2005)]; (b) Detailed map view of the FEBEX drift for reference only [After Keusen et al. (1989) and from NAGRA Technical Report NTB87-14E].

disposal of heat-emitting nuclear fuel waste. Detailed descriptions of the geological settings of the Aar massif of the Central Alps are given by several authors including Stalder (1964); Wüthrich (1965); Steck and Burri (1971); Schaltegger (1990b,a); Schaltegger and Corfu (1992) and references to further studies are given by Goncalves et al. (2012). Geoscientific studies of the Aar granite have been conducted by a number of agencies including NAGRA and ENRESA and these initiatives are documented in several reports and articles by Amiguet (1985); Pahl et al. (1989); Keusen et al. (1989); Möri et al. (2003); Alonso and Alcoverro (2005); Alonso et al. (2005); Rabung et al. (2012); Bouffier (2015); Garralón et al. (2017); Krietsch et al. (2019). In relation to the FEBEX research experiments, the geological setting of the Grimsel Laboratory contains alternate layers of the Aar granite, transition zones and Granodiorite, separated by Lamprophyres and zones that are subjected to intense shearing. A typical view of the geological setting is shown in Figure 1. During the FEBEX experiments, the Grimsel Laboratory was used to conduct heater experiments where the heaters were encapsulated in bentonitic clay. An extensive program of research was conducted by a series of research groups to validate the THM response of both the bentonitic buffer and the rock mass and the results of the research efforts are documented by Alonso and Alcoverro (2005) and Alonso et al. (2005). The Grimsel granite used in this research investigation was obtained from boreholes PRP16.001 and INJ16.001 located in the southern part of the laboratory, drilled from the AU cavern. These boreholes were drilled as a part of the Grimsel



In-situ Stimulation and Circulation (ISC) project (see the location in Figure 1) that investigated the seismo-hydro-
mechanical response of the rock mass to hydraulic stimulation (Amann et al., 2018; Gischig et al., 2018; Doetsch
et al., 2018; Jalali et al., 2018).

During the geological evolution of the Aar Massif, the strata acquired different mineralogical compositions and
the studies by Schaltegger and Krähenbühl (1990) contain very detailed evaluations of the mineralogical composi-
tions of rocks recovered from the Grimsel and Reuss regions. This information is valuable for estimating the solid
material compressibility of the Grimsel granite and for distinguishing the sample locations. For example, the work of
Jokelainen et al. (2013) provides information on the mineralogical composition of the Grimsel granodiorite and the
study by Missana and Garcia-Gutiérrez (2012) provides the mineralogical composition of the FEBEX granite. The
results reported in these investigations are summarized in Tables 1-4 for completeness.

Table 1: Short Petrographical Descriptions of the Rock Samples Analyzed by Schaltegger and Krähenbühl (1990). Compositions are esti-
mated from thin section, all=allanite, ap=apatite, bio=biotite, cc=calcite, chl=chlorite; ep=epidote; fluo=fluorite; gar=garnet; kfs=K-feldspar;
leuc=leucoxene; op=opaques; plag=plagioclase; ser=sericite; sph=sphene; stilp = stilpnomelane; qtz = quartz; zir = zircon.

| Sample No. | Rock Name | Mesoscopic description | Mineralogical composition |
|---|---|---|---|
| KAW 128 | Northern Border Facies, Gurtnellen granite (Reuss valley) | leucocratic, massiv, coarse-grained granite | 38% qtz, 35% kfs, 25% plag, 2% bio; ap, op, all, zir, gar, sph, ep, stilp, chl; |
| KAW 2213A | Grimsel Granodiorite Grimsel lake (Grimsel) | dark, coarse-grained granite to gra­nodiorite, strongly foliated in most cases, augen texture; abundant dark enclaves | 25% qtz, 25% kfs, 38% plag, 12%; bio; ap, op, sph, all, zir, chl, ep, ser, leuc, cc; plag cumulates |
| KAW 2219 | Central Aar Granite s.s., main fa­cies, Chuenzentennen (Grimsel) | coarse-grained granite with only slight cataclastic deformation | 32% qtz, 29% kfs, 31% plag, 8% bio; ap, op, zir, all, leuc, chl, ser, ep |
| KAW 2220 | Central Aar Granite s.s., leucocratic facies, Hangholz (Grimsel) | medium-grained granite, slightly foliated, occurring as stocks and schlieren within the main facies of the Central Aar Granite s.s. | 34% qtz, 32% kfs, 28% plag, 6% bio; ap, op, zir, all, gar, chl, leuc, ser, ep |
| KAW 2408 | Mittagflue Granite, Tschingel bridge (Grimsel) | leucocratic, massive, coarse-grained granite, analogous to the Northern Border Facies of the Reuss valley | 35% qtz, 35% kfs, 28% plag, 2% bio; ap, zir, gar, all, chl, ep, stilp |
| KAW 2427 | Central Aar Granite s.s., main fa­cies, Gelmerstutz (Grimsel) | coarse-grained, massive granite | main rock-forming minerals as KAW 2219, op, all, sph, zir, ap,ep, ser |
| KAW 2518 | Central Aar Granite s.l., Göschenen (Reuss valley) | leucocratic, medium-grained gran­ite, massive to slightly foliated | 32% qtz, 32% kfs, 32% plag, 4% bio; ap, ep, all, zir, gar, ser, leuc |
| KAW 2519 | Central Aar Granite s.l., Schöllenen (Reuss valley) | dark, coarse-grained granodiorite with moderate foliation, augen tex­ture | 27% qtz, 35% plag, 28% kfs, 10% bio; all, zir, op, ap, sph, ep, leuc, chl |
| KAW 2521 | Central Aar Granite s.l. Schöllenen (Reuss valley) | coarse-grained granodiorite, strongly foliated, similar to KAW 2519 | zir, op, ap, sph, ep, leuc, chl same rock-forming minerals as KAW 2519 |
| KAW 2529 | Kessiturm Aplite, white facies (Grimsel) | fine-grained, aplitic (leucogranitic) intrusion of 200 × 800 m within the Grimsel Granodiorite | 40% qtz, 35% kfs, 24% plag, 1% bio; zir, gar, op, fluo, leuc, chl, ep |
| KAW 2532 | Kessiturm Aplite, grey facies (Grimsel) | fine-grained grey aplite, forming blobs and schlieren within the white Kessiturm aplite | 40% qtz, 30% kfs, 28% plag, 2% bio; gar, chl, ep, ser |

Figure 2 shows cores of the Grimsel granite and, from a visual perspective, the rock has the appearance of strati-
fications that would point to the likely presence of transverse isotropy, in terms of its elasticity properties, fluid flow
and fracture and failure characteristics.

The microstructure includes larger crystals of quartz (with dimensions up to 8 mm) and this requires that a suitable
representative volume element is considered, both in the mechanical testing and mineralogical property evaluations.


Table 2: Geochemical Descriptions of the Rock Samples Across the Grimsel Test Site given by Keusen et al. (1989).

| | Central Aare granite | | | | | Grimsel-Granodiorite | | | | | |
|---|---|---|---|---|---|---|---|---|---|---|---|
| | SB1 | SB2 | SB2 | SB3 | SB4 | SB5 | SB5 | SB6 | SB6 | SB6 | SB5 |
| | 74.98 | 14.00 | 74.00 | 93.00 | 72.20 | 35.96 | 39.20 | 48.98 | 59.00 | 75.97 | 39.20 |
| | Wt.% | Wt.% | Wt.% | Wt.% | Wt.% | Wt.% | Wt.% | Wt.% | Wt.% | Wt.% | Wt.% |
| $SiO_2$ | 74.65 | 69.56 | 74.67 | 68.65 | 71.22 | 67.95 | 67.76 | 69.9 | 65.35 | 66.57 | 66.66 |
| $TiO_2$ | 0.2 | 0.41 | 0.16 | 0.42 | 0.41 | 0.58 | 0.61 | 0.44 | 0.51 | 0.56 | 0.47 |
| $Al_2O_3$ | 13.14 | 14.72 | 12.78 | 15.21 | 13.88 | 15.04 | 15.2 | 14.48 | 17.03 | 16.1 | 14.73 |
| $Fe_2O_3$ | 1.39 | 2.98 | 1.13 | 2.97 | 2.6 | 3.44 | 3.58 | 2.71 | 3.3 | 3.61 | 4.1 |
| MnO | 0.04 | 0.1 | 0.04 | 0.09 | 0.07 | 0.07 | 0.07 | 0.06 | 0.08 | 0.08 | 0.09 |
| MgO | 0.24 | 0.69 | 0.18 | 0.69 | 0.56 | 1.27 | 0.54 | 0.76 | 0.91 | 0.88 | 0.12 |
| CaO | 1.01 | 2.08 | 0.93 | 1.97 | 1.84 | 1.85 | 1.29 | 1.71 | 2.56 | 2.83 | 6.99 |
| $Na_2O$ | 3.88 | 4.52 | 3.69 | 4.59 | 3.87 | 4.01 | 4.57 | 3.98 | 4.9 | 4.84 | 3.95 |
| $K_2O$ | 4.7 | 3.47 | 4.83 | 4.03 | 3.91 | 4.03 | 3.77 | 4.59 | 3.56 | 3.35 | 1.57 |
| $P_2O_3$ | 0.07 | 0.13 | 0.05 | 0.13 | 0.12 | 0.19 | 0.19 | 0.14 | 0.16 | 0.18 | 0.15 |
| $Cr_2O_3$ | < 0.01 | < 0.01 | < 0.01 | < 0.01 | < 0.01 | < 0.01 | < 0.01 | < 0.01 | < 0.01 | < 0.01 | < 0.01 |
| NiO | 0.01 | 0.01 | 0.01 | 0.01 | 0.01 | 0.01 | 0.01 | 0.01 | 0.01 | 0.01 | 0.01 |
| Loss of ign. | 0.31 | 0.49 | 0.38 | 0.6 | 0.53 | 0.81 | 0.78 | 0.45 | 0.84 | 0.6 | 0.69 |
| Ignition | 98.92 | 99.15 | 98.84 | 99.35 | 99.01 | 99.24 | 99.36 | 99.22 | 99.2 | 99.6 | 99.52 |

Table 3: Mineralogical composition of the Grimsel Granodiorite (Gr-Gr) [After Jokelainen et al. (2013)].

| Mineral | Sample 1 (Volume %) | Sample 2 (Volume %) |
|---|---|---|
| Plagioclase | 39.0 | 34.0 |
| Quartz | 28.4 | 37.2 |
| K-Feldspar | 21.6 | 12.8 |
| Biotite | 5.0 | 7.8 |
| Muscovite + sericite | 2.6 | 1.6 |
| Epidote | 1.2 | 1.0 |
| Amphibole | 1.8 | 4.6 |
| Chlorite | 0.2 | 0.4 |
| Titanate | - | 0.6 |
| Opaque minerals | 0.2 | - |

Extensive geomechanical characterization studies have been performed on the Grimsel granite and these are given
in the references cited previously. Permeability studies are also reported by Schild et al. (2001). A comprehensive
inter-laboratory study of permeability of the Grimsel granodiorite is also given in David et al. (2018a,b).
The objective of this study is to employ the existing data on the mechanical characterization of the transversely
isotropic granite to estimate the skeletal compressibility of the granite and to use XRD studies of the mineralogical
composition of the Grimsel granite to estimate the compressibility of the solid phase composing the porous fabric.

## 101 2. Skeletal Bulk Modulus of the Grimsel Granite

The fabric of the Grimsel granite is indicative of a transversely isotropic material (Nejati, 2018; Dutler et al., 2018;
Dambly et al., 2019; Nejati et al., 2019). The elastic stress-strain relationships for a transversely isotropic material
can be expressed in several forms (see e.g. Hearmon, 1961; Lekhnitskii, 1963; Ting, 1996). We consider the case
where the plane of isotropy $(x, y)$ of the transversely isotropic elastic material is normal to the $z$-axis. The equations
of elasticity governing the normal strains can be written in the forms



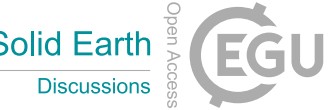

Table 4: Mineralogical composition of the FEBEX Granite [After Missana and Garcia-Gutiérrez (2012)].

| Mineral | Volume (%) |
|---|---|
| Quartz | 30-36 |
| Plagioclase/Albite | 19-23 |
| K-Feldspar | 31-37 |
| Biotite-Chlorite | 6-8 |
| Muscovite | 1-2 |

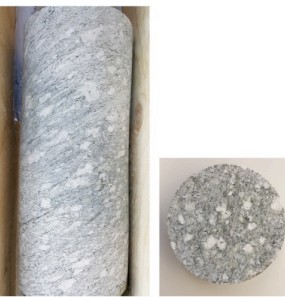

Figure 2: The Grimsel granite cores.

$$\epsilon_{xx} = \frac{\sigma_{xx}}{E_x} - \frac{\nu_{yx}\sigma_{yy}}{E_y} - \frac{\nu_{zx}\sigma_{zz}}{E_z}$$
$$\epsilon_{yy} = -\frac{\nu_{xy}\sigma_{xx}}{E_x} + \frac{\sigma_{yy}}{E_y} - \frac{\nu_{zy}\sigma_{zz}}{E_z} \qquad (1)$$
$$\epsilon_{zz} = -\frac{\nu_{xz}\sigma_{xx}}{E_x} - \frac{\nu_{yz}\sigma_{yy}}{E_y} + \frac{\sigma_{zz}}{E_z}$$

We point out that the Poisson's ratio is generally defined as $\nu_{ij} = -\epsilon_j/\epsilon_i$ for a stress in the $i$ direction. From Betti's
reciprocal theorem,

$$\frac{\nu_{xy}}{E_x} = \frac{\nu_{yx}}{E_y}, \qquad \frac{\nu_{xz}}{E_x} = \frac{\nu_{zx}}{E_z}, \qquad \frac{\nu_{yz}}{E_y} = \frac{\nu_{zy}}{E_z} \qquad (2)$$

Due to the isotropic behaviour in the $xy$ plane, $E_x = E_y$, and $\nu_{zy} = \nu_{zx}$. These relations reduce the independent
material constants needed to define the principal strains to four: $E_x$, $E_z$, $\nu_{xy}$ and $\nu_{zx}$. Consider the situation where an
element of the transversely isotropic elastic medium is subjected to an isotropic compressive stress state: $\sigma_{xx} = \sigma_{yy} =$
$\sigma_{zz} = p$. The infinitesimal volumetric strain

$$\epsilon_v = \epsilon_{xx} + \epsilon_{yy} + \epsilon_{zz} = p\left[\frac{2}{E_x}(1 - \nu_{xy}) + \frac{1}{E_z}(1 - 4\nu_{zx})\right] \qquad (3)$$

The skeletal bulk modulus for the transversely isotropic elastic material can be expressed in the form

$$K_D^{\text{TI}} = \frac{p}{\epsilon_v} = \frac{E_x E_z}{2E_z(1 - \nu_{xy}) + E_x(1 - 4\nu_{zx})} \qquad (4)$$

In terms of the elasticity parameters that are applicable to the direction normal to the planes of stratification ($N$)
and directions along the planes of foliation or stratification ($T$), Eq. (4) can be written as


$$K_D^{\text{TI}} = \frac{E_T E_N}{2E_N(1 - \nu_{TT}) + E_T(1 - 4\nu_{NT})} \tag{5}$$

In the limit of material isotropy, $E_N = E_T = E$ and $\nu_{TT} = \nu_{NT} = \nu$ and Eq. (5) reduces to the classical result

$$K_D^{\text{I}} = \frac{E}{3(1 - 2\nu)} \tag{6}$$

The estimation of the skeletal bulk modulus of the Grimsel granite can be attempted provided that the elasticity
constants applicable to either an isotropic fabric or a transversely isotropic skeletal elastic behaviour, can be identified.
The geomechanical investigations of the granitic rocks at Grimsel have ranged from the estimation of the deformability
and strength characteristics of the rock to the assessment of the in situ stress state. The interpretation of the available
data for estimating the *skeletal deformability characteristics* is complicated by the fact that the approaches used are
not uniform and standardized; the earlier experimental studies may have deviated from currently acceptable standards
(as suggested by ASTM and ISRM) for sample size, rate of loading, end restraints, method of interpretation of
the experimental data for parameter extraction (secant modulus, tangent modulus, loading/unloading paths, cycles),
etc. The exercise is also compounded by the material variability in terms of the Grimsel lithology and its influence
on parameter variability. Within these limitations, attempts can be made to extract, from the existing literature,
representative values of the elasticity characteristics of Grimsel granite with due consideration for the species of
granite. The earliest record used in this study relates to the work of Amiguet (1985) and Alonso and Alcoverro
(2005), which indicate the elasticity properties as $E \approx 60$ GPa; $\nu \approx 0.25$.

Pahl et al. (1989) used borehole dilatometer and overcoring to estimate the in-situ stress state and the overall
deformability characteristics of the granite: $E \approx 40$ GPa; $\nu \approx 0.25$. The work of Keusen et al. (1989) gives a range
of elasticity values applicable to the granodiorite ($max[E \approx 63$ GPa; $\nu \approx 0.48]$; $min[E \approx 32$ GPa; $\nu \approx 0.18]$) and
the Aar granite ($max[E \approx 64$ GPa; $\nu \approx 0.49]$; $min[E \approx 42$ GPa; $\nu \approx 0.25]$). Ziegler and Amann (2012) also report
the results of an extensive series of tests conducted on both wet and dry and coarse-grained and fine-grained samples
of Grimsel granite. The results are presented as maximum and minimum values as follows: for the coarse-grained
granite, $max[E \approx 59$ GPa; $\nu \approx 0.37]$; $min[E \approx 53$ GPa; $\nu \approx 0.25]$; for the medium-grained granite. The recent work
of Bouffier (2015) uses laboratory over coring techniques to estimate the deformability characteristics of the Grimsel
granite and there is a wide range of results for both the elastic modulus and Poisson's ratio; average representative
results are indicated by $E \approx 25$ GPa; $\nu \approx 0.33$. The work of Kant et al. (2017) is primarily focused on the estimation
of the thermal properties of the Aar granite. The results they cite for the modulus of elasticity and Poisson's ratio are
directly obtained from the work of Alonso et al. (2005) or indirectly from Keusen et al. (1989). Wenning et al. (2018)
report studies of permeabilty and seismic velocity anisotropy across a ductile to brittle transition zone in the Grimsel
granite.

The skeletal compressibility is also an important parameter in the interpretation of transient hydraulic pulse tests
for estimating the fluid transport properties of low permeability materials including granite and argillaceous limestones
(Brace et al., 1968; Selvadurai and Carnaffan, 1997; Selvadurai and Selvadurai, 2014; Selvadurai and Najari, 2015).
The elasticity properties were determined via dynamic measurements and the maximum and minimum values are as
follows: $max[E \approx 95$ GPa; $\nu \approx 0.18]$; $min[E \approx 65$ GPa; $\nu \approx 0.15]$. Considering the nature of the ductile to brittle
transmission zone under investigation and the dynamic nature of the tests, these estimates are far in excess of those
for the intact material that is tested statically. Furthermore, the bulk modulus estimated from the maximum values of
$E$ and $\nu$ is in the range of 50 GPa, which is lower than the bulk modulus of mono-mineralic albite but exceeds that of





quartz. The study by Krietsch et al. (2019) deals with the characterization of the in situ stress state at the Grimsel test

site, using a range of experiments including overcoring and hydraulic fracturing. The investigations were extended to

include transverse isotropy of the rock mass.

The elasticity parameters were inferred through a computational back analysis of the overcoring technique; these

authors also provide a comparison with the results obtained by Bouffier (2015). An averaging procedure gives max-

imum estimates of the isotropic elasticity parameters as $E \approx 26$ GPa; $\nu \approx 0.33$. The use of the Grimsel Laboratory

facility for the FEBEX experiment (Alonso and Alcoverro, 2005) provided a useful International Benchmarking ex-

ercise to validate THM modelling of clay buffer regions that could be used in high-level nuclear waste management

endeavours. The international collaborative effort (Alonso et al., 2005) focused more on the behaviour of the clay bar-

rier during heating from emplaced heaters and fluid influx from the Grimsel granite. In many of the research efforts

for the FEBEX Project, the Grimsel granite served as a heat sink and the rock mechanics aspects perhaps received

less emphasis (i.e. the modelling of the bentonitic clay under heating was considered to be the major objective of

the research as opposed to the modelling of the Grimsel granite). Also, to enhance fluid influx, the Grimsel gallery

was considered to be a fractured rock mass and modelling the Grimsel rock elasticity properties varied between the

research groups participating in the FEBEX project, with very low estimates of the elasticity properties (Nguyen et al.,

2005) to near intact rock properties derived from the original studies of Amiguet (1985) (see also Gens et al., 1998;

Alonso and Alcoverro, 2005; Rutqvist et al., 2003; Dupray et al., 2013). For this reason, the elasticity properties of

the Grimsel granite cited in the papers dealing with the FEBEX exercise are excluded from consideration.

The majority of the studies focusing on the evaluation of the deformability characteristics of the Grimsel granite

deal with isotropic elastic modelling. The possible influences of either elastic anisotropy or elastic transverse isotropy

were addressed in the earlier study by Pahl et al. (1989) in connection with the estimation of in situ stress states. In

this particular study, there is no clear statement of the applicable value of the elasticity constants governing transverse

isotropy of the Grimsel granite (the degree of anisotropy $(E_T/E_N)$ does not exceed 1.25) and the study culminates

in the adoption of the isotropic elasticity properties that were indicated previously. The research by Nejati (2018)

and Nejati et al. (2019) deals with the estimation of the deformability characteristics of the Grimsel granite based

on the transversely isotropic elastic model with principal directions aligned in the stratification planes and normal to

the planes (Figure 2). These studies indicate that the Grimsel granite tested also exhibited significant anisotropy and

nonlinearity. In addition, due to nonlinear effects, the secant, tangent and average values of the Young's modulus can

depend on the stress level at which the value is estimated.

If a range of elastic behaviour can be clearly defined and if the elastic constants governing transverse isotropy can

be determined, then, as shown by Eq. (5), the bulk modulus applicable to the transversely isotropic material can be

evaluated objectively. The studies conducted by Nejati (2018) and Nejati et al. (2019) provide the following estimates

for the elastic constants governing the transversely isotropic elasticity model for the Grimsel granite: $E_N \approx 30$ GPa;

$E_T \approx 47$ GPa; $\nu_{TT} \approx 0.20$ GPa; $\nu_{NT} \approx 0.10$ GPa, Finally, Krietsch et al. (2019) conducted a series of experiments on

the ISC core plugs, using overcoring and external pressurization of the hollow samples. These authors also give results

of uniaxial tests conducted on core plugs extracted either normal or parallel to the foliations (Figure 18 of their paper).

These results can be used to estimate the $E_N$ and $E_T$. From the results presented by Krietsch et al. (2019), the relevant

elastic moduli can be summarized as follows: $E_N \approx 13$ GPa; $E_T \approx 35$ GPa. These investigations, however, cannot be

used to estimate the values of $\nu_{TT}$ and $\nu_{NT}$. Dambly et al. (2019) presented the results of a research program geared to

estimate the transversely isotropic elasticity parameters from results of ultrasonic dynamic tests and static tests. Nejati

et al. (2019) compared the static and dynamic values of the elastic constants at zero-confinement, and concluded that





the dynamic moduli are significantly greater than the static ones. In this study we have not considered experimental results derived from dynamic testing; therefore, for consistency, any results derived from dynamic testing of the Grimsel granite have been excluded from further consideration. Considering the experimental evaluations available in the literature, the elasticity parameters applicable to the Grimsel granite are summarized in Table 5.

Table 5: Elasticity Properties for the Grimsel Granite with the corresponding $K_D^I$ or $K_D^{TI}$ values: $K_D^I = E/3(1 - 2\nu)$, $K_D^{TI} = E_T E_N/[2E_N(1 - \nu_{TT}) + E_T(1 - 4\nu_{NT})]$; $N$ signifies the direction normal to the planes of stratification and $T$ signifies the directions along the planes of stratification.

| Reference | Elasticity Type | Elastic Constants | $K_D^I$ or $K_D^{TI}$ |
|---|---|---|---|
| Amiguet (1985)[1] | Isotropic | $E = 60$ GPa; $\nu = 0.25$ | $K_D^I \approx 40$ GPa |
| Pahl et al. (1989) | Isotropic | $E = 40$ GPa; $\nu = 0.25$ | $K_D^I \approx 27$ GPa |
| Keusen et al. (1989) (Granodiorite) | Isotropic | mean $E \approx 47$ GPa; $\nu \approx 0.33$ | $(K_D^I)_{mean} \approx 46$ GPa |
| Keusen et al. (1989) (Aar granite) | Isotropic | mean $E \approx 53$ GPa; $\nu \approx 0.37$ | $(K_D^I)_{mean} \approx 68$ GPa |
| Ziegler and Amann (2012) Type 1– coarse grained | Isotropic | mean $E \approx 38$ GPa; $\nu \approx 0.36$ | $(K_D^I)_{mean} \approx 45$ GPa |
| Ziegler and Amann (2012) Type 2– medium grained | Isotropic | mean $E \approx 43$ GPa; $\nu \approx 0.37$ | $(K_D^I)_{mean} \approx 55$ GPa |
| Bouffier (2015) | Isotropic | $E = 26$ GPa; $\nu = 0.33$ | $K_D^I \approx 25$ GPa |
| Dambly et al. (2019)[1] | Isotropic | $E = 44$ GPa; $\nu = 0.2$ | $K_D^I \approx 24$ GPa |
| Krietsch et al. (2019)[2] | Transversely Isotropic | $E_N \approx 13$ GPa; $E_T \approx 35$ GPa; $\nu_{TT} \approx 0.15$; $\nu_{NT} \approx 0.15$ | $K_D^{TI} \approx 13$ GPa |
| Nejati et al. (2019); Nejati (2018)[3] | Transversely Isotropic | $E_N \approx 30$ GPa; $E_T \approx 47$ GPa; $\nu_{TT} \approx 0.2$; $\nu_{NT} \approx 0.1$ | $K_D^{TI} \approx 19$ GPa |

## 3. Compressibility of the Solid Material Composing the Grimsel Granite Fabric

The skeletal material of the Grimsel granite consists of a variety of mineral phases including quartz, biotite, anorthite, augite, microcline and traces of pyrite and magnetite. The composition of these minerals were determined both at the XRD facilities at University of Montréal, QC, Canada and at the Department of Earth Sciences, Institute of Geology, ETH, Zurich (Wenning et al., 2018). The estimated volume fractions and the values for the bulk moduli and shear moduli are shown in Tables 6 and 7 respectively. The average volume fractions and the mineralogical compositions tend to vary and the estimated values are, in general, considered to be approximate. The results of the XRD evaluations do not provide sufficient accuracy to group the tested rocks into either the Grimsel granodiorite or the FEBEX Grimsel categories. A very cursory comparison with the data provided in Tables 1 to 3 would suggest that the mineralogical compositions provided by Wenning et al. (2018) and indicated in Table 6 correspond to the Grimsel granodiorite and the results shown in Table 7 correspond to the FEBEX granite. For this reason, the XRD data derived from both laboratory evaluations (ETH and McGill) are retained in the estimations of the solid material compressibility $K_S$. Also, the void fraction ($<< 1\%$) is neglected in the calculations. The values for the bulk moduli and shear moduli for the various minerals were obtained from published literature (Alexandrov et al., 1964; Anderson and Nafe, 1965; Carmichael, 1990; Sisodia and Verma, 1990; Moos et al., 1997; Redfern and Angel, 1999; Schilling et al., 2003; Zhu et al., 2007; Mavko et al., 2009; Lin, 2013).

In the multi-phasic approach, the objective is the determine the overall bulk modulus for the solid mineralogical phase by considering the bulk moduli for the separate mineral constituents and their volume fractions. Ideally this

---

[1]This estimate is based on the elastic constants measured along the foliations.

[2]This estimate is based on the secant elastic constants at a stress level of approximately 9 MPa.

[3]This estimate is based on tangent elastic constants at the in-situ stress level based on the analyses of Krietsch et al. (2019), which is approximately 11 MPa.


Table 6: Mineralogical Fractions of the Grimsel Granite [Data obtained by Wenning et al. (2018), Institute of Geology, ETH, Zurich].

| Mineral | Specific Gravity | % | $K_S$ (GPa) | $G_S$ (GPa) |
|---|---|---|---|---|
| Biotite & Phlogopite | 2.72 | 10 | 77 | 42 |
| Muscovite | 2.70 | 5 | 61 | 41 |
| Epidote | 2.75 | 6 | 107 | 60 |
| Albite | 3.19 | 40 | 76 | 26 |
| Feldspar | 2.60 | 16 | 76 | 26 |
| Quartz | 2.72 | 23 | 38 | 45 |
| | | Σ 100 | | |

Table 7: Mineralogical Fractions of the Grimsel Granite [Data obtained by the Earth Sciences Laboratory, University of Montréal].

| Mineral | Specific Gravity | % | $K_S$ (GPa) | $G_S$ (GPa) |
|---|---|---|---|---|
| Quartz | 2.72 | 46 | 38 | 45 |
| Biotite | 2.70 | 5 | 77 | 42 |
| Anorthite | 2.75 | 37 | 68 | 38 |
| Augite | 3.19 | 5 | 95 | 59 |
| Microcline | 2.60 | 7 | 52 | 36 |
| | | Σ 100 | | |

needs to be approached using a generalized theory of multi-phasic composites that can accommodate a mixture of any
number of phases. Such a generalized theory is yet to be developed. The most widely used relationships are those by
Voigt (1928) and Reuss (1929). The Voigt ($^V$) and the Reuss ($^R$) estimates are

$$(K_S)_I^V = \sum_i^n V_i (K_S)_i, \qquad (K_S)_I^R = \left[ \sum_i^n \frac{V_i}{(K_S)_i} \right]^{-1}$$
$$(G_S)_I^V = \sum_i^n V_i (G_S)_i, \qquad (G_S)_I^R = \left[ \sum_i^n \frac{V_i}{(G_S)_i} \right]^{-1} \qquad (7)$$
$$i = Qrtz, Biotite, Anorthite, Augite, Microcline, Voids$$
$$I = \text{Data from Table 1 or Table 2}$$

The results given in Hill (1952, 1965) are the mean of the Voigt and Reuss estimates. This basic approach can be
applied to estimate the effective bulk and shear moduli for the Grimsel granite: i.e.

$$(K_S)_I = \frac{1}{2} \left[ (K_S)_I^V + (K_S)_I^R \right], \qquad (G_S)_I = \frac{1}{2} \left[ (G_S)_I^V + (G_S)_I^R \right] \qquad (8)$$
$$I = \text{Data from Table 1 or Table 2}$$

Using the mineralogical compositions obtained from XRD analyses given in Table 1, we have

$$(K_S)_1 = 65 \text{ GPa}, \qquad (G_S)_1 = 33 \text{ GPa} \qquad (9)$$

and using the mineralogical compositions obtained from XRD analyses given in Table 2, we have

$$(K_S)_2 = 52 \text{ GPa}, \qquad (G_S)_2 = 48 \text{ GPa} \qquad (10)$$

A further approach is to use the Voigt and Reuss estimates for $(K_S)_I$ and $(G_S)_I, (I = 1, 2)$ in the Hashin and
Shtrikman (1963) results to develop bounds for the compressibility of the solid constituents of the Grimsel granite.
The Hashin-Shtrikman results in conjunction with the Voigt and Reuss estimates can be evaluated to estimate the upper
and lower bounds $(K_S)_I^U$ and $(K_S)_I^L, (I = 1, 2)$ for the compressibility of the solid material of the Grimsel granite.





Assuming that a fraction $\phi$ of the Grimsel granite will satisfy the Voigt estimate (or the Reuss estimate), the lower $()^L$ and upper $()^U$ Hashin-Shtrikman bounds for the compressibility of the solid phase, represented by $(K_S)^L$ and $(K_S)^U$, can be obtained from the results

$$(K_S)_I^L = (K_S)_I^V + \frac{\phi}{\dfrac{1}{(K_S)_I^R - (K_S)_I^V} + \left(\dfrac{3(1-\phi)}{3(K_S)_I^V + 4(G_S)_I^V}\right)}, \quad (I = 1, 2) \tag{11}$$

and

$$(K_S)_I^U = (K_S)_I^R + \frac{1 - \phi}{\dfrac{1}{(K_S)_I^V - (K_S)_I^R} + \left(\dfrac{3\phi}{3(K_S)_I^R + 4(G_S)_I^R}\right)}, \quad (I = 1, 2) \tag{12}$$

These bounds converge to the proper limits as $\phi \to 1$ and $\phi \to 0$. The unknown in Eqs. (11) and (12) relates to the volume fraction $\phi$ that will be relevant to the partitioning of fractions of the multi-phasic system that will obey the Voigt and Reuss estimates. There is no physical principle that can be adopted to determine this partitioning, *a priori*. In order to provide a comparison to the Voigt-Reuss-Hill estimate, we can evaluate the expressions (11) and (12) for $\phi = 1/2$, since the Hill estimate is the mean of the Voigt and Reuss estimates. The results for the Hashin-Rosen estimates derived from (11) and (12) give

$$(K_S)_1^L \approx 54.4\,\text{GPa}, \quad (K_S)_2^L \approx 68.5\,\text{GPa} \tag{13}$$

and

$$(K_S)_1^U \approx 49.6\,\text{GPa}, \quad (K_S)_2^U \approx 62.1\,\text{GPa} \tag{14}$$

Considering the limits of the Hashin-Shrtikman estimates for the upper and lower bounds for the solid material compressibility of the Grimsel granite, the *average estimates* for the solid material compressibilities obtained from the two sets of laboratory data give

$$(K_S)_1 \in (49.6, 54.4)\,\text{GPa}, \quad (K_S)_2 \in (62.1, 68.5)\,\text{GPa} \tag{15}$$

Considering the range of solid material compressibilities obtained from the two laboratory investigations we can conclude that the lower $(^L)$ and upper $(^U)$ estimates for $K_S$ are approximately

$$K_S^L \approx 50\,\text{GPa}, \quad K_S^U \approx 69\,\text{GPa} \tag{16}$$

The results for the skeltal compressibilities given in Table 5 can be combined with the range of solid material compressibilities to estimate the *upper* and *lower* limits of the Biot coefficient applicable to each estimate of $K_D^{\text{I}}$ and $K_D^{\text{TI}}$. The relevant results are shown in Table 8.

## 4. Discussion

In theories developed for estimating the elasticity of multi-phasic materials, the most extensive studies relate to two-component elastic materials. Theories, however, have also been developed by several researchers to include a





Table 8: Upper and lower limits for the Biot coefficient for the Grimsel Granite; $\alpha_U = 1 - (K_D^{\mathrm{I}} \text{ or } K_D^{\mathrm{TI}})/K_S^U$, $\alpha_L = 1 - (K_D^{\mathrm{I}} \text{ or } K_D^{\mathrm{TI}})/K_S^L$, $K_S^L \approx 50\,\mathrm{GPa}$, $K_S^U \approx 69\,\mathrm{GPa}$.

| Reference | Elasticity Type | $K_D^{\mathrm{I}}$ or $K_D^{\mathrm{TI}}$ | $\alpha_L$ | $\alpha_U$ |
|---|---|---|---|---|
| Amiguet (1985) | Isotropic | $K_D^{\mathrm{I}} \approx 40\,\mathrm{GPa}$ | 0.19 | 0.42 |
| Pahl et al. (1989) | Isotropic | $K_D^{\mathrm{I}} \approx 27\,\mathrm{GPa}$ | 0.46 | 0.61 |
| Keusen et al. (1989) (Granodiorite) | Isotropic | $(K_D^{\mathrm{I}})_{mean} \approx 46\,\mathrm{GPa}$ | 0.07 | 0.33 |
| Keusen et al. (1989) (Aar granite) | Isotropic | $(K_D^{\mathrm{I}})_{mean} \approx 68\,\mathrm{GPa}$ | -0.37 | 0.01 |
| Ziegler and Amann (2012) Type 1– coarse grained | Isotropic | $(K_D^{\mathrm{I}})_{mean} \approx 45\,\mathrm{GPa}$ | 0.09 | 0.34 |
| Ziegler and Amann (2012) Type 2– medium grained | Isotropic | $(K_D^{\mathrm{I}})_{mean} \approx 55\,\mathrm{GPa}$ | -0.10 | 0.20 |
| Bouffier (2015) | Isotropic | $K_D^{\mathrm{I}} \approx 25\,\mathrm{GPa}$ | 0.50 | 0.64 |
| Dambly et al. (2019) | Isotropic | $K_D^{\mathrm{I}} \approx 24\,\mathrm{GPa}$ | 0.52 | 0.65 |
| Krietsch et al. (2019) | Transversely Isotropic | $K_D^{\mathrm{TI}} \approx 13\,\mathrm{GPa}$ | 0.74 | 0.81 |
| Nejati et al. (2019); Nejati (2018) | Transversely Isotropic | $K_D^{\mathrm{TI}} \approx 19\,\mathrm{GPa}$ | 0.62 | 0.72 |

distribution of three elastic phases in the composite material. An early study in this area is by Cohen and Ishai (1967)
that considered the presence of a large voids content in the two-phase system. Several other developments have been
proposed in the literature; references to such studies are given by Ju and Chen (1994a,b) and the references cited in
the introduction. The extension to three elastic phases was also presented in the studies by Talbot et al. (1995) and,
more recently, by Lin and Ju (2009). Even with these developments, the number of separate components included in
the composite material models are insufficient to accommodate all the components of the solid phases listed in Tables
1 to 3 and 5 and 6.

A plausible alternate approach is to essentially reduce the components in Tables 1 and 2 to three phases by com-
bining (using the Voigt-Reuss-Hill approach) the material phases that correlate closely in terms of their bulk and shear
moduli values. Whether, in view of the approximate nature of the XRD evaluations of the volume fractions of the
separate phases, such refinements are altogether warranted is debatable. The results of the evaluations presented in the
paper suggest that the multi-phasic approach in conjunction with XRD data provides a useful alternative to validating
the conventional experimental approach for estimating the solid material composing low permeability porous media.
The skeletal bulk moduli for the Grimsel granite shows a wide variation, indicative of variable lithology of the igneous
rock formation. In this sense it is prudent to assume a set of limits for the choice of the Biot coefficient rather than
to assign a specific value. Certain data obtained in this study give rise to non-realistic values of the Biot coefficient,
clearly arising from the estimation of the skeletal compressibility.

As a guide, experimental results for the skeletal compressibility values that exceed the effective solid material
compressibility of the minerals with the largest volume fractions should be disregarded. Therefore these results can
be excluded without further comment. (i.e. Since the multi-phasic assessment of the compressibility of the solid
material has a lower limit of approximately $K_S^L \approx 50\,\mathrm{GPa}$, plausible values of the Biot coefficient will be obtained
when $K_D < K_S$.) Also, excessively low values of $K_D$ need to be re-examined before using the data to estimate the
Biot coefficient. Excessively low values can result from inaccurate estimation of the elastic modulus and Poisson's
ratio. Similarly, excessively high values of the skeletal stiffness can result from inaccurate estimates of the Poisson's
ratio of the rock. For example, if samples are loaded in the direction of the foliations or stratifications, micro-crack
or defect development during compression can give rise to lateral deformations that can be a result of void/crack
generation and not a result of material deformation. Considering the numerical values presented in Table 8, and the
above comments, several estimates for the Biot coefficients can be excluded from further discussion. The Table 9
summarizes the revised set of realistic experimental estimates for the Biot coefficient of the Grimsel granite, taking



²⁷⁷ into consideration the aforementioned caveats on the experimental results.

Table 9: Reduced Data Set for the Upper and Lower Limits for the Biot coefficient for the Grimsel Granite.

| Reference | Elasticity Type | $K_D^{\mathrm{I}}$ or $K_D^{\mathrm{TI}}$ | $\alpha_L$ | $\alpha_U$ |
|---|---|---|---|---|
| Pahl et al. (1989) | Isotropic | $K_D^{\mathrm{I}} \approx 27\,\mathrm{GPa}$ | 0.46 | 0.61 |
| Bouffier (2015) | Isotropic | $K_D^{\mathrm{I}} \approx 25\,\mathrm{GPa}$ | 0.50 | 0.64 |
| Dambly et al. (2019) | Isotropic | $K_D^{\mathrm{I}} \approx 24\,\mathrm{GPa}$ | 0.52 | 0.65 |
| Nejati et al. (2019); Nejati (2018) | Transversely Isotropic | $K_D^{\mathrm{TI}} \approx 19\,\mathrm{GPa}$ | 0.62 | 0.72 |

## ²⁷⁸ 5. Conclusions

²⁷⁹ The accurate estimation of the skeletal deformability characteristics of a porous rock is an essential pre-requisite
²⁸⁰ for estimating the Biot coefficient for a fluid-saturated poroelastic material. While the procedures for conducting either
²⁸¹ uniaxial or triaxial tests to estimate the skeletal deformability characteristics are well known, the exact procedure
²⁸² for estimating the elastic moduli, Poisson's ratio, etc., needs to be better documented so that the interpretations of
²⁸³ experimental data can be consistent. The conventional procedure for the pressurization of a saturated sample of
²⁸⁴ the rock and the measurement of the resulting sample strains when the externally applied cell pressure matches the
²⁸⁵ pore fluid pressure is perhaps the best procedure for estimating the compressibility of the solid phases of the porous
²⁸⁶ medium. This, however, is not a routine procedure for low permeability materials and substantial pressures need to be
²⁸⁷ applied to ensure that volumetric strains of an accurately measurable value can be recorded.

²⁸⁸ Also, in such cases the strains could involve irreversible grain boundary frictional slip and this needs to be excluded
²⁸⁹ from the estimation of the solid material compressibility. Here, we advocate the use of a multi-phasic approach where
²⁹⁰ the theories of composite materials can be used to estimate the compressibility of the solid material composing the
²⁹¹ porous skeleton. This is a relatively easy approach since XRD evaluations of the mineralogical phase composition are
²⁹² usually carried out to characterize the rock. In relation to the Grimsel granite, the analysis points to a Biot coefficient
²⁹³ that has bounds rather than a specific value: i.e. $0.46 < \alpha < 0.72$. Values for the Biot coefficient for other types of
²⁹⁴ rocks include [see also Table 1 in Detournay and Cheng (1993)]: Westerly granite ($\alpha \approx 0.47$); for the Lac du Bonnet
²⁹⁵ granite in Manitoba, Canada, a value of $\alpha = 0.73$ is cited (Lau and Chandler, 2004); Ruhr sandstone ($\alpha \approx 0.65$);
²⁹⁶ Berea sandstone ($\alpha \approx 0.79$); Weber sandstone ($\alpha \approx 0.64$); Ohio sandstone ($\alpha \approx 0.65$); Pecos sandstone ($\alpha \approx 0.83$);
²⁹⁷ Boise sandstone ($\alpha \approx 0.85$); Cobourg limestone ($\alpha \approx 0.66$). With soft rocks such as chalk, the Biot coefficient is
²⁹⁸ invariably in the range 0.80 to 1.0 (Alam et al., 2010; Nermoen et al., 2013). For the Callovo-Oxfordian claystone the
²⁹⁹ Biot coefficient is estimated to be in the range of 0.84 (Belmokhtar et al., 2018). Other estimates for a variety of rocks
³⁰⁰ encountered in a coal mining setting are also given by Chen et al. (2019).

## ³⁰¹ Acknowledgement

³⁰² The work described in the paper was supported by a Discovery Research Grant awarded by the Natural Sciences
³⁰³ and Engineering Research Council of Canada. This study is part of the In situ Stimulation and Circulation (ISC)
³⁰⁴ project established by the Swiss Competence Center for Energy Research-Supply of Electricity (SCCER-SoE) with
³⁰⁵ the support of Innosuisse. The authors are also grateful to Professor Eduardo Alonso (UPC, Spain), Professor Lyesse
³⁰⁶ Laloui (EPFL, Switzerland), Professor Florian Amman (RWTH, Germany), Professor Martin Mazurek (University
³⁰⁷ of Bern, Switzerland), Dr. Stratis Vomvoris (NAGRA, Switzerland), Professor Christian David (Université Cergy-
³⁰⁸ Pontoise, France), Dr. Jonny Rutqvist (LBNL, USA), Dr. Farid Laouafa (INERIS, France) and Dr. Son Nguyen





(CNSC, Canada) for drawing attention to the literature used in this study and for helpful comments. The authors gratefully acknowledge the comments made by Dr. Joseph Doetsch, Institute of Geophysics, Department of Earth Sciences ETH Zurich, which led to improvements in the presentation. The authors, however, are entirely responsible for the statements and conclusions presented in the paper.

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

Report, Ingenieurgeologie, ETH Zurich.