# Peer review of "A Multi-phasic Approach for Estimating the Biot Coefficient for Grimsel Granite"

_Solid Earth, 2019_

## Referee Comment (RC1) · Anonymous Referee #1 · 2 Jul 2019

This paper proposes a new method to measure the Biot coefficient of a polymineralic anisotropic rock, namely the Grimsel Gneiss. The paper is well structured and perfectly written. The outcomes of the test result analysis are very convincing and valuable.

---

## Referee Comment (RC2) · Anonymous Referee #2 · 30 Jul 2019

General comments

The measurement of the poroelastic properties of low permeability rocks is not an easy task due to saturation issues and pore pressure artifacts. This paper presents a mixed approach for estimating Biot's coefficient based on direct measurements of the bulk modulus of the porous skeleton and estimations of the bulk modulus of the solid matrix from the sample mineral content and the bulk moduli of the constitutive minerals. The paper is interesting and well-written. The authors first summarize the measures of the elastic constants of Grimsel granite available in the literature and discuss their consistency. The transversely isotropic measurements are used to compute an equivalent isotropic skeletal bulk modulus. The multiphasic approach used to assess the bulk modulus of the solid matrix (Ks) is then detailed and upper and lower bounds are

computed. The obtained bounds are then used to compute upper and lower limits for Grimsel Granite Biot's coefficient. The consistency of the obtained variation ranges is finally discussed and a set of realistic estimates is provided.

Specific comments

Measurement of Biot's coefficient

Additional information on the test system and the followed loading path associated to the "conventional" measurement of Ks considered by the authors would make it easier to follow the reasoning. Is this an unjacketed test? If not, the flow through the cylindrical sample would be linked to its length or half-length according to the applied hydraulic conditions at the sample ends and not to its diameter. Is the sample initially saturated at a given small confining pressure and then both the confining pressure and the pore pressure are simultaneously increased at the same rate? Or is the sample saturated at a higher confining pressure which is then kept constant as the pore pressure is gradually increased? The later approach would allow the checking of the pore pressure equilibrium at each pore pressure increment (see for example Bemer E., Longuemare P., Vincké O., 2004, Poroelastic parameters of Meuse/Haute Marne argillites: Effect of loading and saturation states, Applied Clay Science, 26, 359-356).

Transversely isotropic material

Equations (1) correspond to the behavior of a transversely isotropic dry porous skeleton. The description of the poromechanical behavior of this porous medium would require the introduction of two different Biot's coefficients (Coussy O., 2004, Poromechanics, John Wiley and Sons, USA). Equations (1) are used to define a bulk modulus for the transversely isotropic porous skeleton through Equation (4). The computed "TI" bulk modulus is then used to compute an isotropic Biot's coefficient through the expression $\alpha$=1-(Kd/Ks). The transversely isotropic nature of Grimsel granite is thus not fully considered here. This point should be rapidly discussed in the paper.

Measurement of the elastic properties of Grimsel granite

As a general comment on the literature data on the elastic properties of Grimsel granite, the issue of the saturation state of the tested samples should be more thoroughly discussed. The considered elastic properties should be representative of the skeleton behavior (drained or "dry" properties). The skeletal bulk modulus values deduced from measurements performed on "wet" samples could notably be overestimated. The applied confining level would also be a key parameter as the presence of cracks could lead to underestimated skeletal bulk moduli at low confining pressures. The saturation state issue is all the more pregnant for dynamic measurements as dispersion effects could be high for cracked samples and lead to overestimated skeletal bulk moduli.

Multiphasic approach for computing Ks

The void fraction is said to be neglected in the calculations. As the aim of the computation is to derive the bulk modulus of the solid material, the only relevant void fraction should correspond to occluded porosity. It should be clearly specified in the text. Anyway as it is neglected, it should be removed from Equation (7).

Technical corrections

Lines 27-29: Terzaghi's effective stress corresponds to a Biot's coefficient tending towards unity and thus to a case where the bulk modulus of the solid material is small and not large in comparison to the skeletal bulk modulus. Please correct the misprint.

Figure 1: (b) the legend and details of the figure are too small.

Line 81: One extra bracket

Figure 2: Please specify the sample size and its orientation. White lines oriented at 45° seem to be visible on the sample lateral surface.

Line 136: The maximum and minimum values associated to the medium-grained granite are missing in the text.

[Figure]

Line 185: There is a misprint in the unity of Poisson's ratio estimates.

Line 234: Only the Hashin-Shtrikman reference is given in the text, while the keywords include Hashin-Rosen estimates. "Shtrikman" is misprinted line 237.

Equations (13) and (14): The lower (L) estimates are higher than the upper (U) estimates?

Line 242: "skeletal" is misprinted.

―――――――――――――――――――

---

## Author Comment (AC1) · 4 Aug 2019

The authors gratefully acknowledge the supportive comments made by the reviewer.

---

## Author Comment (AC2) · 4 Aug 2019

Anonymous Referee #2 Received and published: 30 July 2019

**Reviewer's Comments:**

**General comments**

The measurement of the poroelastic properties of low permeability rocks is not an easy task due to saturation issues and pore pressure artifacts. This paper presents a mixed approach for estimating Biot's coefficient based on direct measurements of the bulk modulus of the porous skeleton and estimations of the bulk modulus of the solid matrix from the sample mineral content and the bulk moduli of the constitutive minerals.

The paper is interesting and well-written. The authors first summarize the measures of the elastic constants of Grimsel granite available in the literature and discuss their consistency. The transversely isotropic measurements are used to compute an equivalent isotropic skeletal bulk modulus. The multiphasic approach used to assess the bulk modulus of the solid matrix (Ks) is then detailed and upper and lower bounds are C1 computed. The obtained bounds are then used to compute upper and lower limits for Grimsel Granite Biot's coefficient. The consistency of the obtained variation ranges is finally discussed and a set of realistic estimates is provided.

**Authors' Reply**

The authors are grateful to the reviewer for the constructive and supportive comments. The recommendations of the reviewer will be taken into consideration in the revision of the manuscript.

**Specific comments**

**Reviewer Comment 2.1 Measurement of Biot's coefficient**

Additional information on the test system and the followed loading path associated to the "conventional" measurement of  $K_s$  considered by the authors would make it easier to follow the reasoning. Is this an unjacketed test? If not, the flow through the cylindrical sample would be linked to its length or half-length according to the applied hydraulic conditions at the sample ends and not to its diameter. Is the sample initially saturated at a given small confining pressure and then both the confining pressure and the pore pressure are simultaneously increased at the same rate? Or is the sample saturated at a higher confining pressure which is then kept constant as the pore pressure is gradually increased? The later approach would allow the checking of the pore pressure equilibrium at each pore pressure increment (see for example Bemer E., Longuemare P., Vincké O., 2004, Poroelastic parameters of Meuse/Haute Marne argillites: Effect of loading and saturation states, Applied Clay Science, 26, 359-356).

**Authors' Reply to 2.1**

The ideal arrangement for the measurement of the Ks would involve an unjacketed specimen where the saturating fluid is identical to the pressurizing fluid. In situations where the saturating fluid is water and the pressurizing fluid is oil (needed to attain high pressures without compressibility issues), the sample needs to be jacketed. The sample can be subjected to a constant high confining pressure and the pore fluid pressure increased to attain equilibrium. This appears to be the preferred mode of estimation of the solid material compressibility. Other variations on this procedure are possible depending on the permeability of the rock under investigation. With very low permeability rocks there are issues that need to be recognized in the excessive time for attainment of equalization. The multiphasic approach advocated in the paper stems from this fact. The reference relating to the measurement of poroelastic parameters for the Meuse/Haute Marne argillites is valuable and will be included in the revised version of the paper. If ever there is a criticism in the use of oedometric compression tests for estimating the Biot coefficient, this relates to the radial stress developed in the sample, which is a function of the skeletal Poisson's ratio, which adds a level of uncertainty in the interpretation of the solid material compressibility. In the opinion of the authors, the most appropriate technique is the use of isotropic compression of a jacketed sample. Also, the Meuse/Haute Marne argillite is a clay rock, which will have irreversible deformations in terms of the stress history and the interpretation of the skeletal elasticity properties should reflect stress history. In the case of the Grimsel granite, such effects are not expected to be significant.

**Reviewer Comment 2.2 Transversely isotropic material**

Equations (1) correspond to the behavior of a transversely isotropic dry porous skeleton. The description of the poromechanical behavior of this porous medium would require the introduction of two different Biot's coefficients (Coussy O., 2004, Poromechanics, John Wiley and Sons, USA).

**Authors' Reply to 2.2**

The issue of requiring two different Biot coefficient to describe a single pore pressure variation is not a rational approach. The same applies to formulations that assign a tensorial structure define the Biot coefficient, which would imply that the pore pressure is not a scalar. This is possible only if the fluid pressure at a point is allowed to have different directional values. The extension of Biot's definition to incorporate anisotropy of the fabric is most conveniently done by evaluating the volume change in an anisotropic fabric under isotropic compression.

**Reviewer Comment 2.3**

Equations (1) are used to define a bulk modulus for the transversely isotropic porous skeleton through Equation (4). The computed "TI" bulk modulus is then used to compute an isotropic Biot's coefficient through the expression  $\alpha$ =1-(KD/KS). The transversely isotropic nature of Grimsel granite is thus not fully considered here. This point should be rapidly discussed in the paper. Measurement of the elastic properties of Grimsel granite As a general comment on the literature data on the elastic properties of Grimsel granite, the issue of the saturation state of the tested samples should be more thoroughly discussed. The considered elastic properties should be representative of the skeleton behavior (drained or "dry" properties). The skeletal bulk modulus values deduced from measurements performed on "wet" samples could notably be overestimated. The applied confining level would also be a key parameter as the presence of cracks could lead to underestimated skeletal bulk moduli at low confining pressures. The saturation state issue is all the more pregnant for dynamic measurements as dispersion effects could be high for cracked samples and lead to overestimated skeletal bulk moduli. Multiphasic approach for computing Ks. The void fraction is said to be neglected in the calculations. As the aim of the computation is to derive the bulk modulus of the solid material, the only relevant void fraction should correspond to occluded porosity. It should be clearly specified in the text. Anyway as it is neglected, it should be removed from Equation (7).

**Authors' Reply to 2.3**

The approach adopted in the paper is to consider a transversely isotropic dry Grimsel rock, whose transversely isotropic elastic constants have been determined through separate tests. Once these parameters are available, a hypothetical sample of the dry Grimsel granite can be subjected to an isotropic state of stress and this will result in the development of normal strains in the sample without distortion. The principal strains can be calculated and if infinitesimal strains are considered, the volumetric strain is the first invariant of the infinitesimal strain tensor. The reviewer is correct in outlining other issues related to the testing of "wet samples" and "testing at high confining stresses" that could induce crack closure/crack opening, which can influence the estimation of elasticity parameters and subsequently the estimation of KD. While these issues can be addressed in relation to the experiments performed by the authors, the task of determining the exact test conditions that other researchers followed is not so straightforward. The authors can therefore only adopt the values reported in the literature. The concluding remarks in the paper emphasizes the need to be vigilant about test procedures, particularly when reporting values for the elasticity properties.

**Reviewer Comment 2.4 Technical corrections**

(a) Lines 27-29: Terzaghi's effective stress corresponds to a Biot's coefficient tending towards unity and thus to a case where the bulk modulus of the solid material is small and not large in comparison to the skeletal bulk modulus. Please correct the misprint.

Figure 1: (b) the legend and details of the figure are too small.

Line 81: One extra bracket

Figure 2: Please specify the sample size and its orientation. White lines oriented at 45\_seem to be visible on the sample lateral surface.

Line 136: The maximum and minimum values associated to the medium-grained granite are missing in the text. C3

Line 185: There is a misprint in the unity of Poisson's ratio estimates.

Line 234: Only the Hashin-Shtrikman reference is given in the text, while the keywords include Hashin-Rosen estimates. "Shtrikman" is misprinted line 237.

Equations (13) and (14): The lower (L) estimates are higher than the upper (U) estimates?

Line 242: "skeletal" is misprinted.

Authors' Reply to 2.4 The authors greatly appreciate the Technical Comments made by the reviewer and, where relevant, these will be corrected/implemented in the revised version of the paper.

---

## Author Response (AR2)

[revised manuscript text omitted]
 8. It may be noted that the composite materials approach outlined here has been used in the geomechanics and geosciences areas and references to these can be found in the volumes by Ahrens (1995); Markov and Preziosi (2000); Mavko et al. (2009) and in the articles by Suvorov and Selvadurai (2011) and Selvadurai (2019).

Table 8: Upper and lower limits for the Biot coefficient for the Grimsel Granite; $\alpha_U = 1 - (K_D^I \text{ or } K_D^{TI})/K_S^U$, $\alpha_L = 1 - (K_D^I \text{ or } K_D^{TI})/K_S^L$, $K_S^L \approx 52\,\mathrm{GPa}$, $K_S^U \approx 66\,\mathrm{GPa}$.

| Reference | Elasticity Type | $K_D^I$ or $K_D^{TI}$ | $\alpha_L$ | $\alpha_U$ |
|---|---|---|---|---|
| Amiguet (1985) | Isotropic | $K_D^I \approx 40\,\mathrm{GPa}$ | 0.23 | 0.39 |
| Pahl et al. (1989) | Isotropic | $K_D^I \approx 27\,\mathrm{GPa}$ | 0.48 | 0.59 |
| Keusen et al. (1989) (Granodiorite) | Isotropic | $(K_D^I)_{mean} \approx 46\,\mathrm{GPa}$ | 0.12 | 0.30 |
| Keusen et al. (1989) (Aar granite) | Isotropic | $(K_D^I)_{mean} \approx 68\,\mathrm{GPa}$ | -0.31 | -0.03 |
| Ziegler and Amann (2012) Type 1– coarse grained | Isotropic | $(K_D^I)_{mean} \approx 45\,\mathrm{GPa}$ | 0.13 | 0.32 |
| Ziegler and Amann (2012) Type 2– medium grained | Isotropic | $(K_D^I)_{mean} \approx 55\,\mathrm{GPa}$ | -0.06 | 0.17 |
| Bouffier (2015) | Isotropic | $K_D^I \approx 25\,\mathrm{GPa}$ | 0.52 | 0.62 |
| Dambly et al. (2019) | Isotropic | $K_D^I \approx 24\,\mathrm{GPa}$ | 0.54 | 0.64 |
| Krietsch et al. (2019) | Transversely Isotropic | $K_D^{TI} \approx 13\,\mathrm{GPa}$ | 0.75 | 0.80 |
| Nejati et al. (2019); Nejati (2018) | Transversely Isotropic | $K_D^{TI} \approx 19\,\mathrm{GPa}$ | 0.63 | 0.71 |

**4. Discussion**

In theories developed for estimating the elasticity of multi-phasic materials, the most extensive studies relate to two-component elastic materials. Theories, however, have also been developed by several researchers to include a distribution of three elastic phases in the composite material. An early study in this area is by Cohen and Ishai (1967) that considered the presence of a large voids content in the two-phase system. Several other developments have been proposed in the literature references to studies are given by Selvadurai (2019) and the other references cited in the introduction. The extension to three elastic phases was also presented in the studies by Talbot et al. (1995) and, more recently, by Lin and Ju (2009).

Here we have used the theoretical estimates proposed by Vogt and Reuss and modified by Hill, and the bounds proposed by Walpole to estimate the upper and lower bound values for the effective bulk moduli of the solid phase. It is shown that the estimates proposed by Voigt-Reuss-Hill and those of Walpole yield practically the same values. The results of the evaluations presented in the paper would suggest that the multiphasic approach in conjunction with XRD data provides a useful alternative to validating the conventional experimental approach for estimating the solid material composing low permeability porous media. The skeletal bulk moduli for the Grimsel granite shows a wide variation, indicative of variable lithology of the igneous rock formation. In this sense, it is prudent to assume a set of limits for the choice of the Biot coefficient rather than to assign a specific value. Certain data obtained in this study give rise to non-realistic values of the Biot coefficient, clearly arising from the estimation of the skeletal compressibility.

As a guide, experimental results for the skeletal compressibility values that exceed the effective solid material compressibility of the minerals with the largest volume fractions should be disregarded. Therefore these results can be excluded without further comment. (i.e. Since the multi-phasic assessment of the compressibility of the solid material has a lower limit of approximately $K_S^L \approx 50\,\mathrm{GPa}$, plausible values of the Biot coefficient will be obtained when $K_D < K_S$.) Also, excessively low values of $K_D$ need to be re-examined before using the data to estimate the Biot coefficient. Excessively low values can result from inaccurate estimation of the elastic modulus and Poisson's ratio. Similarly, excessively high values of the skeletal stiffness can result from inaccurate estimates of the Poisson's ratio of the rock. For example, if samples are loaded in the direction of the foliations or stratifications, micro-crack or defect development during compression can give rise to lateral deformations that can be a result of void/crack generation and not a result of material deformation. Considering the numerical values presented in Table 8, and the above comments, several estimates for the Biot coefficients can be excluded from further discussion. The Table 9

summarizes the revised set of realistic experimental estimates for the Biot coefficient of the Grimsel granite, taking into consideration the aforementioned caveats on the experimental results. Within the context of Biot's theory of poroelasticity, the deformability of the skeletal fabric and the constituent solids is always assumed to be linearly elastic. The developments can also be extended to include elasto-plastic behaviour of the porous skeleton (Suvorov and Selvadurai, 2019).

Table 9: Reduced Data Set for the Upper and Lower Limits for the Biot coefficient for the Grimsel Granite.

| Reference | Elasticity Type | $K_D^{\text{I}}$ or $K_D^{\text{TI}}$ | $\alpha_L$ | $\alpha_U$ |
|---|---|---|---|---|
| Pahl et al. (1989) | Isotropic | $K_D^{\text{I}} \approx 27\,\text{GPa}$ | 0.48 | 0.59 |
| Bouffier (2015) | Isotropic | $K_D^{\text{I}} \approx 25\,\text{GPa}$ | 0.52 | 0.62 |
| Dambly et al. (2019) | Isotropic | $K_D^{\text{I}} \approx 24\,\text{GPa}$ | 0.54 | 0.64 |
| Nejati et al. (2019); Nejati (2018) | Transversely Isotropic | $K_D^{\text{TI}} \approx 19\,\text{GPa}$ | 0.63 | 0.71 |

**5. Conclusions**

The accurate estimation of the skeletal deformability characteristics of a porous rock is an essential pre-requisite for the estimation of the Biot coefficient for a fluid-saturated poroelastic material. While the procedures for conducting the either uniaxial or triaxial tests for estimation the skeletal deformability characteristics is well known, the exact procedure for estimating the elastic moduli, Poisson's ratio, etc., needs to be better documented so that the interpretations of experimental data can be consistent. The conventional procedure for the pressurization of a saturated sample of the rock and the measurement of the resulting sample strains when the externally applied cell pressure matches the pore fluid pressure is perhaps the best procedure for estimating the compressibility of the solid phases of the porous medium. This, however, is not a routine procedure for low permeability materials and substantial pressures need to be applied to ensure that volumetric strains of an accurately measurable value can be recorded. Also, in such cases the strains could involve irreversible grain boundary frictional slip and this needs to be excluded from the estimation of the solid material compressibility.

Here, we advocate the use of a multiphasic approach where the theories of composite materials can be used to estimate the compressibility of the solid material composing the porous skeleton. This is a relatively easy approach since XRD evaluations of the mineralogical phase composition are usually carried out to characterize the rock. In relation to the Grimsel granite, the analysis points to a Biot coefficient that has bounds rather than a specific value: i.e. $0.48 < \alpha < 0.71$. Values for the Biot coefficient for other types of rocks include the following [see also Table 1 in Detournay and Cheng (1993)]: Westerly granite ($\alpha \approx 0.47$). Values for the Biot coefficient for other types of granite in Manitoba, Canada, a value of $\alpha = 0.73$ is cited (Lau and Chandler, 2004); Sandstones have also shown this same variability: Ruhr sandstone ($\alpha \approx 0.65$), Berea sandstone ($\alpha \approx 0.79$), Weber sandstone ($\alpha \approx 0.64$), Ohio ($\alpha \approx 0.65$), Pecos sandstone ($\alpha \approx 0.83$) and Boise sandstone ($\alpha \approx 0.85$) [Further estimates are provided by Zimmerman (1991)]; Cobourg limestone ($\alpha \approx 0.66$). With soft rocks such as chalk, the Biot coefficient is invariably in the range 0.80 to 1.0 (Alam et al., 2010; Nermoen et al., 2013). For the Callovo -Oxfordian claystone the Biot coefficient is estimated to be in the range of 0.84 (Belmokhtar et al., 2018). Biot coefficient for gas-bearing tight sandstone is estimated at $\alpha \approx 0.38$ (Selvadurai, 2019). Other estimates for a variety of rocks encountered in a coal mining setting are also given by Chen et al. (2019).

**Acknowledgement**

The work described in the paper was supported by a Discovery Research Grant awarded by the Natural Sciences and Engineering Research Council of Canada. This study is part of the In situ Stimulation and Circulation (ISC) project established by the Swiss Competence Center for Energy Research-Supply of Electricity (SCCER-SoE) with the support of Innosuisse. The authors are also grateful to Professor Eduardo Alonso (UPC, Spain), Professor Lyesse Laloui (EPFL, Switzerland), Professor Florian Amman (RWTH, Germany), Professor Martin Mazurek (University of Bern, Switzerland), Dr. Stratis Vomvoris (NAGRA, Switzerland), Professor Christian David (Université Cergy-Pontoise, France), Dr. Jonny Rutqvist (LBNL, USA), Dr. Farid Laouafa (INERIS, France) and Dr. Son Nguyen (CNSC, Canada) and Professor R.W. Zimmerman (Imperial College, UK) for helpful comments. The authors grate-fully acknowledge the comments made by Dr. Joseph Doetsch, Institute of Geophysics, Department of Earth Sciences ETH Zurich, which led to improvements in the presentation. The authors, however, are entirely responsible for the statements and conclusions presented in the paper. Finally, the authors are grateful to the reviewers for their highly constructive comments that led to improvements in the presentation.